# Weyl-like points from band inversions of spin-polarised surface states in NbGeSb

I. Marković[1,2], C.A. Hooley [1], O.J. Clark [1], F. Mazzola[1], M.D. Watson [1], J.M. Riley[1], K. Volckaert[1], K. Underwood[1], M.S. Dyer [3], P.A.E. Murgatroyd[4], K.J. Murphy [4], P. Le Fèvre[5], F. Bertran [5], J. Fujii[6], I. Vobornik [6], S. Wu[7], T. Okuda [8], J. Alaria[4] & P.D.C. King [1]*

Band inversions are key to stabilising a variety of novel electronic states in solids, from topological surface states to the formation of symmetry-protected three-dimensional Dirac and Weyl points and nodal-line semimetals. Here, we create a band inversion not of bulk states, but rather between manifolds of surface states. We realise this by aliovalent substitution of Nb for Zr and Sb for S in the ZrSiS family of nonsymmorphic semimetals. Using angle-resolved photoemission and density-functional theory, we show how two pairs of surface states, known from ZrSiS, are driven to intersect each other near the Fermi level in NbGeSb, and to develop pronounced spin splittings. We demonstrate how mirror symmetry leads to protected crossing points in the resulting spin-orbital entangled surface band structure, thereby stabilising surface state analogues of three-dimensional Weyl points. More generally, our observations suggest new opportunities for engineering topologically and symmetry-protected states via band inversions of surface states.

[1] SUPA, School of Physics and Astronomy, University of St Andrews, St Andrews KY16 9SS, United Kingdom. [2] Max Planck Institute for Chemical Physics of Solids, Nöthnitzer Strasse 40, 01187 Dresden, Germany. [3] Department of Chemistry, University of Liverpool, Liverpool L69 7ZD, United Kingdom. [4] Department of Physics, University of Liverpool, Liverpool L69 7ZE, United Kingdom. [5] Synchrotron SOLEIL, CNRS-CEA, L'Orme des Merisiers, Saint-Aubin-BP48, 91192 Gif-sur-Yvette, France. [6] Istituto Officina dei Materiali (IOM)-CNR, Laboratorio TASC, Area Science Park, S.S.14, Km 163.5, 34149 Trieste, Italy. [7] Graduate School of Science, Hiroshima University, 1-3-1 Kagamiyama, Higashi-Hiroshima 739-8526, Japan. [8] Hiroshima Synchrotron Radiation Center, Hiroshima University, 2-313 Kagamiyama, Higashi-Hiroshima 739-0046, Japan. *email: philip.king@st-andrews.ac.uk

The routes by which different underlying symmetries can lead to a variety of topologically protected electronic states in solids have garnered great attention in the past years[1–5]. ZrSiS[6], for example, has recently been established as a bulk Dirac nodal line system[7–16], with multiple lines of Dirac points protected by its nonsymmorphic and mirror crystalline symmetries[4,7,17–19]. When its nonsymmorphic symmetry is broken at the crystal surface, new sets of electronic states are created, split off from the bulk manifold[8]. One such surface state, SS, of dominantly Zr character, intersects the Fermi level ($E_F$), while another, SS′, of predominantly S character, is located at much higher binding energies[8,20]. The existence of these surface states is a result of very general symmetry considerations[8], and so should be expected to be ubiquitous across this materials class. It is still possible to tune their properties, however. For example, isovalent replacement of Zr by the heavier Hf atom leads to a similar set of surface states, but with the Hf-derived state developing a moderate spin splitting as a result of its larger spin-orbit coupling[9,21].

Here, we employ aliovalent substitution of Nb for Zr and Sb for S in order to make a more dramatic modification of the surface state spectrum, ultimately realising a band inversion of the surface states. We demonstrate how the enhanced spin-orbit coupling leads to the development of pronounced Rashba-like spin splittings and show, both experimentally and via ab initio calculations, how this leads to a rich crossing structure of the spin-polarised surface states. We find how oppositely polarised states hybridise where they cross, but exhibit a large and unusual asymmetry in the scale of their hybridisation gaps, while protected, Weyl-like points are formed at the crossings of like-spin bands. We show how this results from the coupled spin-angular and orbital-angular momentum texture of these states, and demonstrate a critical role of mirror symmetry in the surface plane.

## Results

**Spin-split surface states of NbGeSb.** For this work, we have synthesised single crystals of the hitherto little-studied nonsymmorphic compound NbGeSb[22,23] (see Methods section). NbGeSb is isostructural to ZrSiS. As shown in Fig. 1c, it contains square nets of Ge atoms in the $ab$ plane, with two Nb layers and two Sb layers located between each pair of neighbouring Ge planes. The atomic positions within each pair of Nb or Sb layers are related to each other via a combined mirror reflection and translation by a fraction of the unit cell, making the symmetry group of this structure nonsymmorphic. NbGeSb shares the same total charge count as ZrSiS, and should therefore retain charge compensation between electron-like and hole-like carriers[24]. However, Sb has one less valence electron than S, while Nb has one more than Zr. A pronounced change in the relative positions and occupations of the underlying electronic states can thus be expected as compared to ZrSiS. The SS surface state, which forms a small electron pocket around $\overline{X}$ in ZrSiS[7–9], should become more electron-doped in NbGeSb, thus being pushed down in energy. Moreover, SS′, which is located more than 1.5 $e$V below the Fermi level at the $\overline{X}$ point in ZrSiS[7–9], would be expected to become hole-doped, thus being raised up to intersect the Fermi level in the vicinity of $\overline{X}$.

Fig. 1 illustrates how the above expectation is borne out in reality. Comparison of our angle-resolved photoemission (ARPES) measurements (Fig. 1a) with density-functional theory (DFT) supercell calculations (Fig. 1b) allows us to identify both the SS and SS′ manifolds of surface states in NbGeSb. Both are clearly resolved where they disperse through large $k_z$-projected band gaps in the surface Brillouin zone (see also Supplementary Fig. 1). As compared to ZrSiS, the two manifolds of surface states have been strongly shifted towards each other, in fact to such an

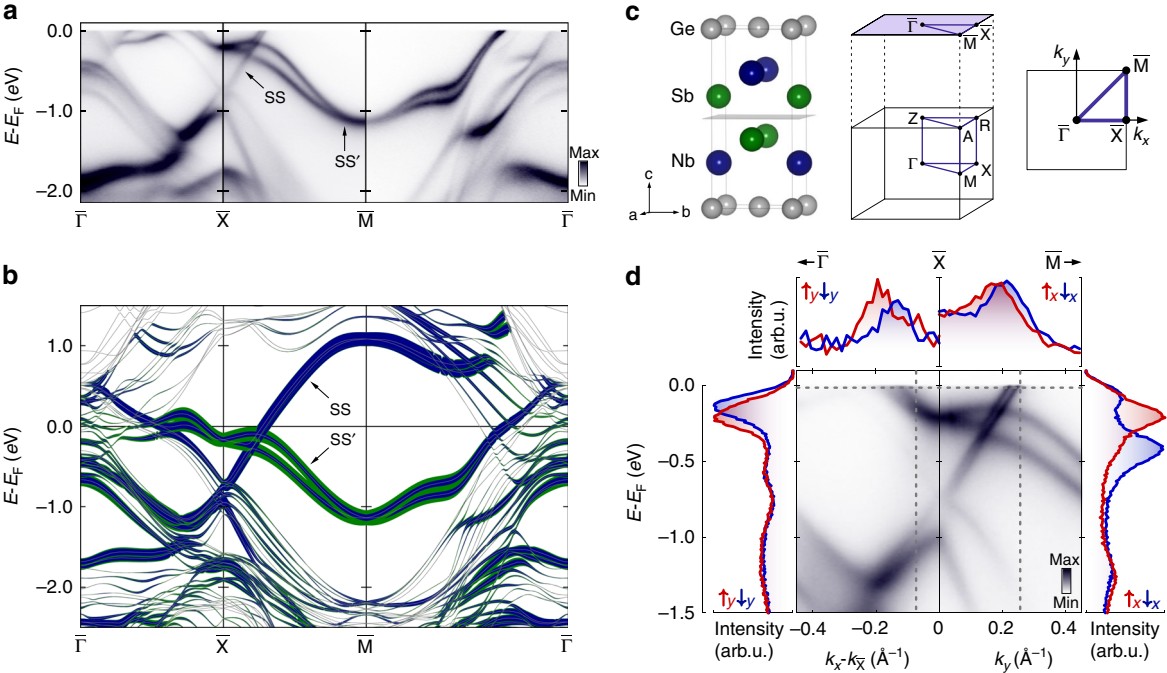

**Fig. 1** Spin-polarised surface electronic structure of NbGeSb. **a** ARPES dispersions along the high-symmetry lines of the surface-projected Brillouin zone of NbGeSb. **b** Corresponding DFT slab calculation, with line colour and weight representing the wavefunction projection onto the surface Nb (blue) and Sb (green) atoms. **c** Crystal structure of NbGeSb with its primary cleavage plane shown; the bulk and surface projected Brillouin zones are also shown. **d** ARPES measurements in the vicinity of the $\overline{X}$ point. A clear splitting of each set of surface states is visible and spin-resolved energy (EDC) and momentum (MDC) distribution curves (shown left/right and top) show this to be a spin splitting. The spin quantisation axis is normal to the high-symmetry line along which the measurement is performed.

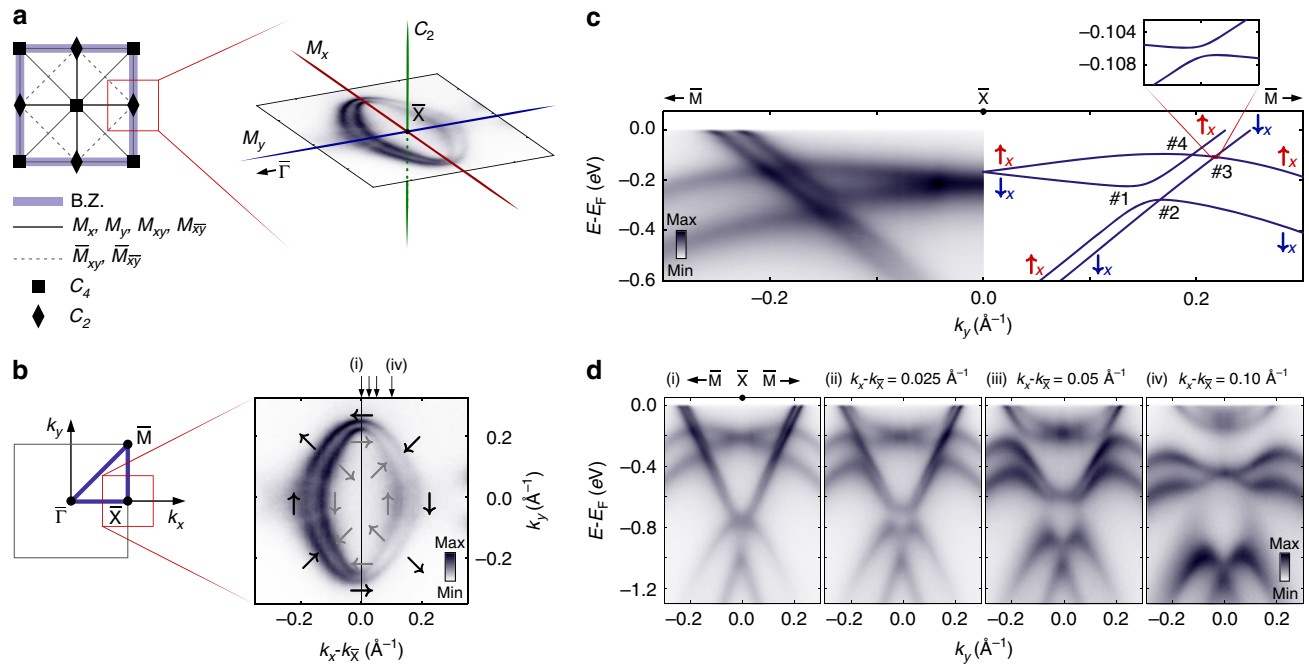

**Fig. 2** Mirror-symmetry protected surface band crossings. **a** Symmetry elements of the *p4mm* layer group[50], which applies to the surface projected Brillouin zone and the surface unit cell of NbGeSb. The inset shows the relevant symmetries at the $\overline{\mathrm{X}}$ point overlaid on the surface state Fermi surfaces measured in the vicinity of this point. **b** The resulting spin textures of these Fermi surfaces are shown schematically by arrows. **c** High-resolution ARPES measurements (left) and DFT calculations (right) in the vicinity of the surface state crossings along $\overline{\mathrm{X}} - \overline{\mathrm{M}}$. The four band crossings are numbered and spin polarisation is again indicated by arrows. Inset is a close-up of crossing #3, showing a small hybridisation gap of ca. 1 meV. **d** ARPES dispersions measured off the $\overline{\mathrm{X}} - \overline{\mathrm{M}}$ line, along the cuts indicated in **b**, demonstrating the evolution of the crossing structure away from the $M_x$ mirror line at the Brillouin zone edge.

extent that they now cross through each other in the vicinity of the Fermi level.

Our measurements and calculations indicate a clear splitting of each surface state into two branches away from the high-symmetry (time-reversal invariant momentum) $\overline{\mathrm{X}}$ and $\overline{\mathrm{M}}$ points. Spin-resolved energy (EDC) and momentum (MDC) distribution curves (Fig. 1d) reveal a clear spin polarisation of these surface states, where the measured spin projection reverses sign between the upper and lower split-off branches of each surface state. We attribute this surface state spin splitting to the Rashba effect[25], whereby spin degeneracy is lifted by spin-orbit coupling when inversion symmetry is broken at the surface. The magnitude of the induced spin splitting is large, reaching up to ∼ 225 meV for the SS′ manifold close to the $\overline{\mathrm{X}}$ point (Fig. 1d).

The corresponding surface state spin texture is also strongly constrained by the high symmetry of the surface unit cell (Fig. 2a). Its $C_4$ rotational symmetry, combined with time-reversal symmetry, immediately precludes any out-of-plane spin canting. Moreover, the $\overline{\Gamma} - \overline{\mathrm{X}}$, $\overline{\mathrm{X}} - \overline{\mathrm{M}}$, and $\overline{\Gamma} - \overline{\mathrm{M}}$ directions are all mirror lines. Along such lines, all non-degenerate states must be eigenstates of the mirror operator, constraining the spin to lie perpendicular to these mirror lines, fully consistent with our experimental measurements (see also Supplementary Fig. 2). Governed by these constraints, the resulting surface state Fermi surface develops a highly structured spin texture, as shown schematically in Fig. 2b.

**Surface state band crossings**. This Fermi surface is already a product of the intertwining of the two sets of surface states: the Fermi crossing along $\overline{\mathrm{X}} - \overline{\Gamma}$ is from the SS′ surface state, while that along $\overline{\mathrm{X}} - \overline{\mathrm{M}}$ is from the SS state (Fig. 1d). Indeed, in

contrast to ZrSiS where the surface state electron pocket smoothly shrinks with increasing binding energy[9], here a complex evolution of the constant energy contours is observed (Supplementary Fig. 3), showing a succession of Lifshitz-like transitions as the surface states pass through each other. Along the high-symmetry $\overline{\mathrm{X}} - \overline{\mathrm{M}}$ line, a quartet of band crossings is formed where the two pairs of surface states disperse through each other (Fig. 2c). Two are between bands which have the same spin polarisation as each other (crossings #2 and #4 in Fig. 2c), and two are between bands with opposite spin polarisation (crossings #1 and #3 in Fig. 2c). Intriguingly, our measurements indicate that there is negligible band hybridisation at three of the crossing points, while crossing #1, between oppositely spin-polarised states, develops a pronounced hybridisation gap on the order of 50 meV. DFT calculations performed on a very dense **k**-grid (inset of Fig. 2c) indicate that the crossing between opposite-spin states that appears to be protected in the experiment (#3) in fact opens a very small band gap. This is, however, on the order of only 1 meV, too small to be resolved experimentally here, and a factor of ca. 50 times smaller than the gap opening at the nearby partner crossing. In contrast, clearly resolvable hybridisation gaps open at all four crossing points when off the high-symmetry line (Fig. 2d), indicating a critical role of the mirror symmetry which is present along the Brillouin zone boundary (Fig. 2a) in generating the structure observed here.

The strongly asymmetric nature of the hybridisation structure along the mirror line is highly unusual: for a conventional bulk state where the orbital degree of freedom is quenched by the crystal field, there is no a priori reason to distinguish between the $(\downarrow_x, \uparrow_x)$ and $(\uparrow_x, \downarrow_x)$ crossings, and comparable hybridisation matrix elements would be expected at both. The presence of inversion symmetry breaking at the surface, however, allows

additional orbital hybridisations which can drive the development of unquenched orbital angular momentum[26–30].

Along the mirror line, the surface eigenstates must have definite mirror parity[17,31]. This is evident in our DFT slab calculations shown with the wavefunction weight projected onto atomic orbital basis states of the surface layer atoms (Supplementary Fig. 4). Along $\overline{X} - \overline{M}$, the SS band consists only of orbitals that are even under $M_x$, which form the basis for the orbitally unquenched states with $L_x = \{-2, 0, +2\}$. In contrast, the SS′ band is composed only of orbitals which are odd under $M_x$, forming the basis for $L_x = \{-1, +1\}$. The OAM states of the two bands thus belong to orthogonal manifolds. Further insight into the OAM structure can be gained from the magnitude of the spin splitting. In particular, the maximum size of the observed spin splittings here ($\sim 90$ meV for SS and $\sim 225$ meV for SS′, Fig. 2c) are comparable to the average of the atomic spin-orbit coupling of Nb $4d$ and Sb $5p$ orbitals[32,33] weighted by their relative contribution to the band. This points to the energy scale associated with inversion symmetry breaking being greater than the atomic spin-orbit coupling strength[30]. As such, a pronounced OAM can be expected here, with approximately the same expectation value for each of the spin-split branches of a given surface state[28–30].

**Orbital angular momentum.** To explore this further, we construct a simple tight-binding model based on a symmetry analysis of the allowed inter-orbital hopping terms for the NbGeSb surface structure (see Methods section). The full band dispersions from this model are shown in Supplementary Fig. 5, while we focus here on the $\overline{X} - \overline{M}$ direction (Fig. 3a) for which a similar structure of two protected and two asymmetrically gapped crossings arises to that observed experimentally for NbGeSb. Fig. 3b shows the resulting OAM which develops for these surface bands. Just as for the spin angular momenta, mirror symmetry constrains the OAM to point perpendicular to the mirror line; along the $M_x$ mirror line shown here, only the perpendicular OAM component, $L_x$, can therefore develop. In line with the above considerations based on the magnitude of the spin splitting observed experimentally, both spin-split branches of the SS′ surface state develop a similar OAM, which we find is very close to a pure $L_x = 1$ state (Fig. 3b). Both branches of the SS surface state are close to $L_x = 0$, although the small, but non-zero, expectation value of the OAM indicates that this band additionally hosts some admixture of $L_x = \pm 2$.

As a good starting approximation, we neglect the small admixture of $L_x = \pm 2$ and consider the SS and SS′ bands to be purely in neighbouring levels of the underlying OAM manifold (i.e., $L_x = 0$ and $L_x = 1$, respectively). The resulting minimal model of the four surface band crossings is that of a two-level

system for both spin and orbital angular momenta, as shown schematically in Fig. 3(c). Along the mirror lines, only atomic spin-orbit coupling of the $\mathbf{L} \cdot \mathbf{S}$ form can open a hybridisation gap at the band crossings, as inter-orbital mixing between the states of opposite mirror parity is otherwise strictly forbidden. The atomic spin-orbit coupling is naturally represented in the $x$ basis as $\mathbf{L} \cdot \mathbf{S} = L_x S_x + \frac{1}{2}(L_+ S_- + L_- S_+)$[34], due to the pinning of both spin and orbital angular momenta perpendicular to the mirror line.

The "spin-flip" terms (last two terms) connect states for which the angular momentum change of the spin is opposite to that of the orbital sector. This is the situation only for crossing #1 here, which thus opens a clear hybridisation gap, as observed for the real material (Fig. 2c). In contrast, at crossing #3, between $|L_x = 1, \uparrow_x\rangle$ and $|L_x = 0, \downarrow_x\rangle$ states, the spin-flip terms cannot act because the angular momentum change of the spin is of the same sign as that of the orbital sector. Moreover, the $L_x S_x$ term cannot act at any of the crossings because the states have different $L_x$ projection at all of these. Within our minimal model, crossing #3 is therefore protected. In reality, the admixture of $L_x = \pm 2$ into SS removes this strict protection by allowing a contribution from the spin-flip terms. The resulting hybridisation gap will, however, naturally be much smaller than at crossing #1, explaining the highly asymmetric nature of the hybridisation gap structure observed for NbGeSb. In the tight-binding example shown in Fig. 3a, the gap at crossing #3 is 14 times smaller than that at crossing #1, while an even larger ratio is found for the real material (Fig. 2c), indicating that deviations from the effective two-level system employed in Fig. 3c are small in reality.

At the like-spin crossings, #2 and #4, the spin-flip terms of the atomic spin-orbit coupling cannot act because the spin projection in each band is the same, while the $L_x S_x$ term again cannot act because the states have different $L_x$ projection. This remains true even including deviations from pure OAM eigenstates, because the spin is still polarised along $x$ for states along the mirror line, and the orbital angular momentum states of the two bands belong to strictly orthogonal manifolds. These crossings are thus strictly protected by the defined mirror parity of the bands.

**Generation of Weyl-like points.** Moving off the Brillouin zone boundary, inter-orbital mixing and spin canting become allowed, and so the crossing points along the mirror line represent isolated band degeneracies. Fig. 4a shows the resulting band dispersions in the vicinity of one of these protected crossings (#4). This has the characteristic form of a type-I tilted Weyl cone known from three-dimensional solids[35], but now confined to a two-dimensional surface layer. Conventionally, stabilising Weyl points is assumed to require three dimensions in order to tune all

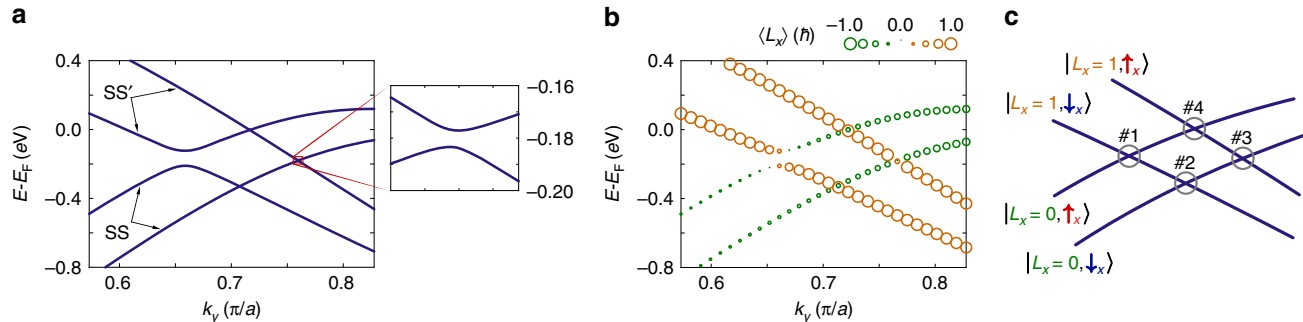

**Fig. 3** Orbital angular momentum of the surface states. **a** The fourfold crossing of the SS and SS′ surface states in our tight-binding model (see Methods section). A magnified view of crossing #3 is shown inset. **b** Reproduction of the tight-binding dispersions from **a**, with the calculated expectation value of the orbital angular momentum along the $x$ quantisation axis shown as symbol colour/size. **c** Schematic of the fourfold crossing of SS and SS′ surface states. The crossings are numbered as in Fig. 2c, and ket labels on the bands represent the orbital and spin angular momenta of the bands in our minimal model.

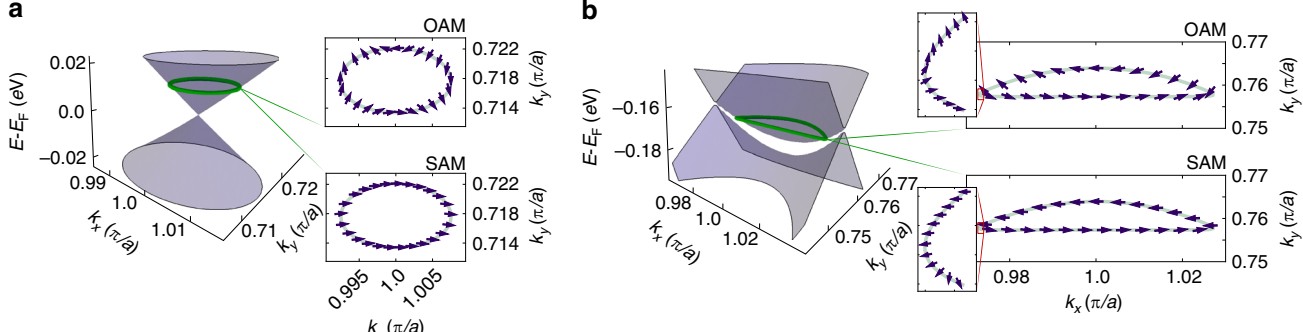

**Fig. 4** Weyl-like points in the surface band structure of NbGeSb. **a** Band dispersions around the like-spin protected crossing (#4 from Fig. 3c) as calculated from our tight-binding model. Orbital (top) and spin (bottom) angular momenta extracted around the indicated contour are shown as insets; the arrows represent the direction of the angular momenta. The OAM shows a characteristic winding around the contour with winding number −1, unlike for the spin which exhibits only slight canting around the contour. **b** Equivalent calculations around the weakly gapped crossing (#3 in Fig. 3c). Now both the orbital (top inset) and spin (bottom inset) angular momenta exhibit winding around the indicated contour, with winding numbers −1 and +1, respectively, but the majority of the winding is restricted in close proximity to the mirror line, as can be seen in the magnified insets.

hybridisation terms to zero: in two dimensions, a perturbation of $\sigma_z$ type would thus generically be expected to open a band gap in the quasiparticle spectrum. Here, the mirror symmetry at the Brillouin zone edge provides a key additional symmetry constraint, preventing such hybridisations and allowing the realisation of Weyl-like points from band inversions of two-dimensional surface states. This is reminiscent of the situation in black phosphorus, where surface doping with alkali metals was recently shown to modify the bulk band structure so as to drive a band inversion in the near-surface region, with Dirac states stabilised by glide-mirror symmetry[36]. Here, however, rather than arising via surface modification of the bulk states, it is the inherent surface states of the material which themselves exhibit an intrinsic band inversion. Moreover, a strong spin-orbit coupling means that the band crossings occur between spin-polarised states, as identified above, leading to the generation of Weyl-like rather than Dirac-like states.

At the protected crossings, the two bands which intersect have the same spin polarisation as each other, but different orbital angular momentum parity. In contrast to a conventional Weyl cone, the spin texture here thus exhibits no winding, but rather only a weak canting around the cone (Fig. 4a–bottom right). Instead, it is the orbital angular momentum that winds, in order to interpolate between the two opposite parities as one goes round a small loop in **k**-space around the crossing point (Fig. 4a–top right). The chiral pseudospin of the analogue Weyl cones which form is therefore derived from their OAM, with the effective low-energy Hamiltonian given by a $2 \times 2$ matrix of the Weyl form: $H_{\text{eff}} = \nu \, \xi_z (\tau_x p_y + \tau_y p_x)$ (see Methods section), where **p** is the in-plane momentum measured relative to the crossing point, $\nu$ is a velocity, $\boldsymbol{\tau}$ is an orbital pseudospin, and $\xi_z$ is a pseudospin-1/2 variable describing which zone face the crossing is on. We thus denote these crossings as Weyl-like[5]. At the weakly gapped crossing #3, the two bands have opposite spin polarisation, as well as different orbital angular momentum parity. Both the spin and the orbital angular momentum therefore exhibit winding (Fig. 4 (b)), leading to an arguably even richer entanglement of the spin and orbital degrees of freedom. This winding would be present even if the gap induced by the $\mathbf{L} \cdot \mathbf{S}$ spin-orbit term were to close. We note that, because both the SS and SS′ bands are split by the same Rashba part of the spin-orbit coupling, removal of spin-orbit coupling causes all four crossings—the two Weyl-like ones and the two asymmetrically gapped ones—to merge at the same point in reciprocal space.

## Discussion

The above findings illustrate how surface states, as well as bulk states, can provide a rich environment for realising, and indeed extending, the rich variety of electronic structures that can be generated from band inversions in solids, opening new opportunities to profit from the different symmetries and environment of the surface itself. Further theoretical work is needed to ascertain whether the Weyl cone analogues observed here would in principle support long-lived one dimensional edge states. More generally, it may be possible to stabilise such edge states as the boundary modes of parity-inverted surface state band gaps, even if the bulk of the material is itself topologically trivial. This would provide an interesting alternative to hinge-state systems[37–39], where 1D edge modes are also created, but as a result of a higher-dimensional bulk topological invariant.

For NbGeSb studied here, the simultaneous presence of bulk states crossing the Fermi level close to the Brillouin zone centre would complicate the observation of signatures of the surface Weyl-like cones or corresponding edge states in, for example, transport measurements. Nonetheless, the observations from surface-sensitive spectroscopy here motivate the future search for new materials where the protected surface state crossings are tuned into a projected gap of the bulk electronic spectrum, and thus become accessible to transport. In this regard, we note that surface state energies can often be more readily manipulated than those of corresponding bulk states, for example via surface adsorption or even electrical gating of interface states, providing greater opportunities for manipulating the underlying surface electronic structure than are present for bulk states. The concept of surface band inversions outlined here may therefore present a highly tunable platform for inducing and manipulating topological and symmetry-protected surface electronic states of materials.

## Methods

**Sample synthesis**. High-quality NbGeSb single crystals (space group P4/nmm, no. 129[6,22]) were grown via chemical vapour transport using iodine as the transport agent[40]. Elemental Nb (Alfa Aesar 99.99%), Ge (Alfa Aesar 99.999%) and Sb (Alfa Aesar 99.99%) were combined stoichiometrically in the molar ratio 1:1:1 under an argon atmosphere and ground into a fine powder. Dry lumped iodine in the concentration 10 mg/cm³ was then added to the powder and both were sealed in a 20 cm quartz tube under vacuum. The quartz tube was then kept in a two-zone gradient furnace for 336 h with the hot end at 850 °C and the cooler end at 750 °C. After cooling to room temperature, shiny rod-like crystals were obtained at the cooler end and single crystal x-ray diffraction confirmed growth of NbGeSb.

## Angle-resolved photoemission

The samples were cleaved in situ at the measurement temperature of $\sim 10-20$ K. Spin-integrated angle-resolved photoemission (ARPES) measurements were performed at the CASSIOPEE beamline of Soleil synchrotron, France, while spin-resolved ARPES measurements were performed utilising very low-energy electron diffraction (VLEED)-based spin polarimeters[41] at the BL-9B beamline of Hiroshima Synchrotron Radiation Center (HiSOR), Japan[42], and at the APE beamline of Elettra synchrotron, Italy[43]. The Sherman functions were calibrated for the utilised targets via reference measurements of the surface states of Bi(111) and Au(111) for BL-9B and APE, respectively, and the data shown in Fig. 1d and Supplementary Fig. 1 has been normalised to account for this finite Sherman function. The dispersions shown in Fig. 1a and Supplementary Fig. 2 were measured using $h\nu = 65\,eV$ circularly polarised light; the data is shown as the sum of measurements performed using circular-left and circular-right polarised light. All other photoemission data, both spin-integrated and spin-resolved, was measured using $h\nu = 18\,eV$ linearly $p$-polarised light, except for the slices shown in Fig. 2d and Supplementary Fig. 3 which are presented as the sum of measurements performed using $p$-polarised and $s$-polarised light in order to aid visibility of all of the band features.

## Density-functional theory

Electronic band structures were calculated using density functional theory (DFT). Plane-wave based periodic calculations were performed using the VASP programme[44]. Core-electrons were treated using the projector-augmented wave method[45], with the exception of Nb $4s$, $4p$, and Ge $3d$ states which were treated as valence electrons. Calculations used the PBE functional[46] with the inclusion of spin-orbit coupling[47]. All calculations used a plane-wave cutoff energy of 500 $eV$.

The cell parameters and atomic positions of bulk NbGeSb were optimised starting from the experimentally reported crystal structure and using a grid of $20 \times 20 \times 9$ k-points until all forces fell below $0.001\,eV/\text{Å}$. A five unit cell thick slab of NbGeSb was then created by repeating the relaxed bulk structure seven times in the $c$ direction and removing two unit cells such that the slab was symmetrically terminated by layers of Sb atoms. All atomic coordinates were subsequently optimised within a fixed periodic cell, using a $20 \times 20 \times 1$ k-point grid, until all forces were reduced to below $0.001\,eV/\text{Å}$. This generated a slab of NbGeSb with a vacuum region of 18.36 Å separating the Sb surface layers in periodic images. Subsequent calculations of the electronic structure at specific k-points were performed non-self-consistently using the electron density generated using the full $20 \times 20 \times 1$ k-point grid.

## Tight-binding calculations

To explore the physics of the surface band structure of NbGeSb in a controllable setting, we created a tight-binding model for the electrons in the five $d$-orbitals of the surface Nb atoms. Including symmetry-allowed mixings of the Sb orbitals would not influence any of the essential understanding here, and so we neglect them for simplicity. The resulting tight-binding model thus has ten basis states per Nb atom: the orbitals $d_{3z^2-r^2}$, $d_{x^2-y^2}$, $d_{yz}$, $d_{xy}$, and $d_{xz}$ with spin-projection up, and the same orbitals with spin-projection down.

We build the Hamiltonian for this model in three stages,

$$H = H_0 + H_R + H_{SO} : \qquad (1)$$

- First, we consider the spin-independent tight-binding hopping processes between nearby Nb atoms, $H_0$. We use only relatively near-neighbour form factors: $c_+ \equiv \cos k_x + \cos k_y$; $c_- \equiv \cos k_x - \cos k_y$; $s_x \equiv i \sin k_x$; $s_y \equiv i \sin k_y$; and $s_{xy} \equiv \sin k_x \sin k_y$. The allowed form factors are constrained by the symmetries of the model: $C_4$ rotational symmetry, the mirror symmetries $M_x$ and $M_y$, and time-reversal symmetry. The $M_z$ mirror is not enforced, reflecting the fact that inversion symmetry is broken along $z$ by the presence of the surface potential. Because the $C_4$ rotational symmetry mixes the $d_{xz}$ and $d_{yz}$ orbitals, it constrains certain elements of $H_0$ to have the same hopping integral as each other. We choose the remaining hopping integrals arbitrarily. The resulting Hamiltonian is

$$H_0 = \begin{pmatrix} \Delta_1 + t_1 c_+ & t_2 c_- & t_3 s_y & t_4 s_{xy} & t_3 s_x \\ t_2 c_- & \Delta_2 + t_5 c_+ & t_6 s_y & 0 & -t_6 s_x \\ -t_3 s_y & -t_6 s_y & \Delta_3 + t_7 c_+ + t_8 c_- & t_9 s_x & t_{10} s_{xy} \\ t_4 s_{xy} & 0 & -t_9 s_x & \Delta_4 + t_{11} c_+ & -t_9 s_y \\ -t_3 s_x & t_6 s_x & t_{10} s_{xy} & t_9 s_y & \Delta_3 + t_7 c_+ - t_8 c_- \end{pmatrix},$$

$$(2)$$

and the parameters we have utilised are $\Delta_1 = \Delta_2 = -1$, $\Delta_3 = \Delta_4 = 0$, $t_1 = -0.5$, $t_2 = -0.75$, $t_3 = t_4 = t_5 = t_6 = -1$, $t_7 = 0.75$, $t_8 = 0.25$, $t_9 = 1$, $t_{10} = 0.3$, and $t_{11} = 0.5$.
- Second, we add Rashba spin-orbit coupling of the form $H_R = \alpha_R \hat{\mathbf{z}} \cdot (\mathbf{S} \times \mathbf{k})$ with $\alpha_R = 0.2$. This splits each band of $H_0$ into a pair of bands with opposite spin orientations, though the axis along which the spins are oriented is different in different points of the surface Brillouin zone.
- Third, we add intra-unit-cell spin-orbit coupling of the form $H_{SO} = \alpha_{SO} \mathbf{L} \cdot \mathbf{S}$ with $\alpha_{SO} = 0.02$. While in reality both $H_R$ and $H_{SO}$ originate from the same microscopic spin-orbit term in the real-space Hamiltonian, it is convenient to

divide that term into an intra-unit-cell component $H_{SO}$ (which is taken to be independent of the crystal momentum $\mathbf{k}$) and an inter-unit-cell component $H_R$[48]. This allows us to investigate the influence of orbital mixing generically allowed by the $\mathbf{L} \cdot \mathbf{S}$ term on the crossings generated by the Rashba term.

The band structure of the resulting model is shown in Supplementary Fig. 5. Near zero energy on the $\overline{X} - \overline{M}$ line it shows a crossing structure similar to that of NbGeSb, as reproduced in Fig. 3a, b of the main text. Notice, in particular, that without fine-tuning of the parameters of the tight-binding model we naturally obtain a very large asymmetry (a factor of around 14) between the size of the small and large gaps arising from the effect of $H_{SO}$ at the unprotected crossings.

## Low-energy effective Hamiltonian

It is clear from Fig. 4 that the orbital angular momentum in the bands has non-trivial winding around the protected crossing, just as the spin does in a conventional Weyl point. Since there are two bands involved, each with an orbital angular momentum that varies only slowly in the vicinity of the protected crossing, we may represent the orbital angular momentum using a pseudospin-1/2 variable $\boldsymbol{\tau}$ which has the same transformation properties under the spatial symmetries and time-reversal as the physical orbital angular momentum $\mathbf{L}$. The corresponding low-energy effective Hamiltonian, valid in the vicinity of the Weyl-like points, is

$$H_{\text{eff}} = \nu\, \xi_z \left( \tau_x p_y + \tau_y p_x \right). \qquad (3)$$

Here $\mathbf{p}$ is the in-plane momentum measured relative to the crossing point, $\nu$ is a velocity, and $\xi_z$ is a pseudospin-1/2 variable describing which zone face the crossing is on: $\xi_z = -1$ for the face at $k_x a = \pi$ and $\xi_z = +1$ for the face at $k_y a = \pi$.

This effective Hamiltonian is invariant under the symmetries of the zone face:

- *Time reversal.* Under time reversal, both $\boldsymbol{\tau}$ and $\mathbf{p}$ change sign, while $\boldsymbol{\xi}$ does not, leaving $H_{\text{eff}}$ unchanged.
- *Mirrors.* Under the $M_x$ mirror operation, $\tau_x$ and $p_y$ are invariant, while $\tau_y$ and $p_x$ change sign. Hence the term in parentheses is invariant overall; since $\xi_z$ is also unchanged by $M_x$, $H_{\text{eff}}$ is invariant under $M_x$. The same argument with $x$ and $y$ interchanged demonstrates the invariance of $H_{\text{eff}}$ under $M_y$.
- $C_4$ *rotation.* $C_4$ rotation makes the following changes:

$$\begin{aligned} p_x &\rightarrow p_y, \quad p_y \rightarrow -p_x, \\ \tau_x &\rightarrow \tau_y, \quad \tau_y \rightarrow -\tau_x, \quad \xi_z \rightarrow -\xi_z. \end{aligned} \qquad (4)$$

Therefore the term in parentheses changes sign, but so does $\xi_z$, meaning that $H_{\text{eff}}$ is invariant overall.

The low-energy effective Hamiltonian in Eq. (3) is written in the 'natural' basis, i.e., one where the orbital pseudospin $\boldsymbol{\tau}$ is aligned with the actual orbital angular momentum $\mathbf{L}$. This makes Eq. (3) look different from the conventional $\boldsymbol{\tau} \cdot \mathbf{p}$ form of the Weyl Hamiltonian. However, under rotation of the pseudospin vector $(\tau_x, \tau_y) \rightarrow (\tau_y, -\tau_x)$ followed by mirror reflection in the $xz$ plane $(\tau_x, \tau_y) \rightarrow (-\tau_x, \tau_y)$, the low-energy effective Hamiltonian becomes

$$H_{\text{eff}} = \nu\, \xi_z \left( \tau_x p_y + \tau_y p_x \right) \quad \rightarrow \quad \nu\, \xi_z \left( \tau_y p_y - \tau_x p_x \right) \quad \rightarrow \quad \nu\, \xi_z \left( \tau_y p_y + \tau_x p_x \right), \qquad (5)$$

which is of the usual Weyl form[5]. Since neither of these operations changes the magnitude of the winding number, these low-energy effective Hamiltonians are equivalent from the topological point of view.

## Data availability

The research data that underpins the findings of this study are available at https://doi.org/10.17630/ae8b005a-f154-4ef8-bc53-e0e89db41023.[49]

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

## Acknowledgements

We thank G. A. Fiete, A. P. Mackenzie, and V. Sunko for enlightening discussions. We gratefully acknowledge The Leverhulme Trust (Grant No. PLP-2015-144), The Royal Society, and the Engineering and Physical Sciences Research Council, UK (Grant No. EP/R031924/1) for support. We are grateful to Soleil Synchrotron for access to beamline CASSIOPEE, Elettra Synchrotron for access to the APE beamline, and HiSOR Synchrotron for access to beamline BL-9B under the HiSOR Proposal No. 17BG022, which all contributed to this work, and the CALIPSOplus project under Grant Agreement 730872 from the EU Framework Programme for Research and Innovation HORIZON 2020. I.M. acknowledges financial support by the International Max Planck Research School for Chemistry and Physics of Quantum Materials (IMPRS-CPQM). O.J.C., J.M.R., and K.U. acknowledge EPSRC for studentship support through grant nos. EP/K503162/1 and EP/L505079/1, and EP/L015110/1. This work has been partly performed in the framework of the Nanoscience Foundry and Fine Analysis (NFFA-MIUR, Italy) facility. C.A.H. is grateful to Rice University for their hospitality during a four-month visiting professorship, where part of this work was carried out.

## Author contributions

The experimental data were measured by I.M., O.J.C., F.M., M.D.W., J.M.R., K.V., K.U., and P.D.C.K., and analysed by I.M. C.A.H. developed the theoretical models in discussions with I.M. and P.D.C.K. M.S.D. and P.A.E.M. performed the DFT calculations. K.J.M. and J.A. synthesised the measured samples. P.L.F., F.B., J.F., I.V., S.W., and T.O. maintained the ARPES/spin-resolved ARPES end stations and provided experimental support. I.M., P.D.C.K., and C.A.H. wrote the paper with input and discussion from co-authors. P.D.C.K. was responsible for overall project planning and direction.

## Competing interests

The authors declare no competing interests.
