## [Peer Review File · Nature Communications]

Reviewers' comments:

Reviewer #1 (Remarks to the Author):

I found that the authors have made appropriate revisions, considering my past comments in full. The revised version of manuscript is improved in many aspects, and I found no reason to further delay its publication in Nature Communications.

Reviewer #2 (Remarks to the Author):

The authors have addressed most of the points, but I am still not convinced by the connection to the Weyl hamiltonian. I am wondering if the low energy effective Hamiltonian, which they present in the methods section, is too simplified. Firstly, the authors do not provide any reference to existing literature and how this connects to the Weyl hamiltonians that are described in the references that the authors provide in the reply (Vafek and Vishvanath as well as Young and Kane). Second, the Weyl hamiltonians that I usually find feature a scalar product of spin and momentum. The hamiltonian in Eq. 3 looks more like a cross product, but not like a scalar product. How do the authors justify this change? Further, the review paper (Vafek and Vishvanath) talks about a Zeeman field that induces the Weyl points or line nodes. How is this carried over to the present case? Along these lines, in Young and Kane, case IV can be reduced to the previous cases (i. e. Dirac points) by restoring spin degeneracy (inversion symmetry). Restoring spin degeneracy in the present case would not move all pairs of Weyl points into one single point, so that I am wondering if the Weyl analogy is appropriate here. This also raises the question, where the 1D analogy of Fermi arcs would appear. I do not believe that they would appear between the Weyl points at opposite momenta. Unless these points are clarified, I cannot recommend publication of this manuscript.

Reviewer #3 (Remarks to the Author):

The authors have answered my previous comments in a satisfactory manner. I recommend this paper for publication in its present form.

Response to Referees

We thank all three referees for reading our revised manuscript. We are pleased that both referee 1 and referee 3 found that their previous comments had been addressed in full and recommend publication of the manuscript in Nature Communications in its present form. Referee 2 raises several related questions to which we are happy to provide further clarification below and in the revised manuscript (blue text):

The authors have addressed most of the points, but I am still not convinced by the connection to the Weyl hamiltonian. I am wondering if the low energy effective Hamiltonian, which they present in the methods section, is too simplified. Firstly, the authors do not provide any reference to existing literature and how this connects to the Weyl hamiltonians that are described in the references that the authors provide in the reply (Vafek and Vishvanath as well as Young and Kane). Second, the Weyl hamiltonians that I usually find feature a scalar product of spin and momentum. The hamiltonian in Eq. 3 looks more like a cross product, but not like a scalar product. How do the authors justify this change?

As the referee points out, the low-energy effective Hamiltonian we present does not appear in the standard Weyl form. This is the case because we have written our orbital pseudospin in the 'natural' basis where it aligns with the orbital angular momentum. However, we are free to choose another definition of the orbital pseudospin basis, through which the Hamiltonian can be brought into the standard Weyl form. Specifically, under rotation of the pseudospin vector $(\tau_x, \tau_y) \rightarrow (\tau_y, -\tau_x)$ followed by mirror reflection in the xz plane $(\tau_x, \tau_y) \rightarrow (-\tau_x, \tau_y)$ the low-energy effective Hamiltonian becomes:

$$H_{\text{eff}} = v \xi_z (\tau_x p_y + \tau_y p_x) \rightarrow v \xi_z (\tau_y p_y - \tau_x p_x) \rightarrow v \xi_z (\tau_y p_y + \tau_x p_x),$$

which is of the usual Weyl form. Since neither of these operations changes the magnitude of the winding number, these low-energy effective Hamiltonians are equivalent from the topological point of view, and thus our low-energy model is indeed of a Weyl-like form. We have added additional text to the 'Methods' section of the manuscript to explain this point, including reference to the Vafek and Vishvanath paper. We are grateful for the suggestion to further clarify this point.

Further, the review paper (Vafek and Vishvanath) talks about a Zeeman field that induces the Weyl points or line nodes. How is this carried over to the present case? Along these lines, in Young and Kane, case IV can be reduced to the previous cases (i. e. Dirac points) by restoring spin degeneracy (inversion symmetry). Restoring spin degeneracy in the present case would not move all pairs of Weyl points into one single point, so that I am wondering if the Weyl analogy is appropriate here.

In NbGeSb, the Rashba spin-orbit coupling plays the role of an effective k-space-local magnetic field, producing locally similar results to that of an applied magnetic field but in a way that globally respects time-reversal symmetry. Restoring inversion symmetry in the present case does in fact merge the two Weyl-like points, but these simultaneously merge

with the two asymmetrically gapped crossings, leaving just a single 4-fold degenerate crossing point. We have now discussed this in the revised text.

This also raises the question, where the 1D analogy of Fermi arcs would appear. I do not believe that they would appear between the Weyl points at opposite momenta.

We agree with the referee that this is a very interesting and important question. However, while it seems likely that the presence of Weyl-like points in the 2D surface Brillouin zone would be associated with special 1D states on the edges of the crystal in real space, it is not obvious to us that the 'Fermi arc' phenomenology established for 3D Weyl systems extends directly to this lower-dimensional case. The Fermi arc is not a state in itself; it is a singular feature in the distribution function, which could be absent here even if non-trivial edge states persist. Recent work by Morice, Kopp, and Kampf (<https://arxiv.org/abs/1909.11782>) has identified significant ambiguities in the bulk-surface mapping even in the usual case, which will have to be properly worked out for NbGeSb before a definite statement can be made. Furthermore, the pseudospin that winds around the Weyl-like points in NbGeSb is associated with the orbital rather than the spin angular momentum of the electrons, and this may alter the relationship between the bulk and surface states. Therefore, we take the view that this question, while clearly interesting, is most suitable as a topic for future work.

Unless these points are clarified, I cannot recommend publication of this manuscript.

We hope that, in light of the abovementioned revisions and our answers to referee 2's questions, that you will now find the manuscript suitable for publication in Nature Communications.

REVIEWERS' COMMENTS:

Reviewer #2 (Remarks to the Author):

The authors have addressed the points, so I can now recommend publication of the manuscript.